# Comparative Transcriptome Analysis of the Molecular Mechanism of the Hairy Roots of *Brassica campestris* L. in Response to Cadmium Stress

**DOI:** 10.3390/ijms21010180

**Published:** 2019-12-26

**Authors:** Yaping Sun, Qianyun Lu, Yushen Cao, Menghua Wang, Xiyu Cheng, Qiong Yan

**Affiliations:** College of Life Sciences and Bioengineering, School of Science, Beijing Jiaotong University, Beijing 100044, China; 17121607@bjtu.edu.cn (Y.S.); 15121580@bjtu.edu.cn (Q.L.); 16121632@bjtu.edu.cn (Y.C.); 18121618@bjtu.edu.cn (M.W.)

**Keywords:** *Brassica campestris* L., cadmium, glutathione synthetase, glutathione *S*-transferase, transcriptome

## Abstract

*Brassica campestris* L., a hyperaccumulator of cadmium (Cd), is considered a candidate plant for efficient phytoremediation. The hairy roots of *Brassica campestris* L are chosen here as a model plant system to investigate the response mechanism of *Brassica campestris* L. to Cd stress. High-throughput sequencing technology is used to identify genes related to Cd tolerance. A total of 2394 differentially expressed genes (DEGs) are identified by RNA-Seq analysis, among which 1564 genes are up-regulated, and 830 genes are down-regulated. Data from the gene ontology (GO) analysis indicate that DEGs are mainly involved in metabolic processes. Glutathione metabolism, in which glutathione synthetase and glutathione *S*-transferase are closely related to Cd stress, is identified in the Kyoto Encyclopedia of Genes and Genomes (KEGG) analysis. A Western blot shows that glutathione synthetase and glutathione *S*-transferase are involved in Cd tolerance. These results provide a preliminary understanding of the Cd tolerance mechanism of *Brassica campestris* L. and are, hence, of particular importance to the future development of an efficient phytoremediation process based on hairy root cultures, genetic modification, and the subsequent regeneration of the whole plant.

## 1. Introduction

Cadmium (Cd) is one of the most toxic and most common heavy metals, causing serious pollution to farmland and active transfer in soil-plant systems [1,2,3,4,5,6,7,8,9,10,11,12]. Cd intake may lead to tissue inflammation, fibrosis, and even various cancers in humans [13,14,15,16,17], necessitating the development of an efficient strategy to eliminate Cd from soil. Phytoremediation is an effective and low-cost technique for removing Cd from contaminated soil [18,19,20]. Currently, many studies in this area have been carried out using whole plants, which have a limited lifespan and must be replaced and re-established after each experiment. It is also difficult to evaluate the effects of various factors such as light, temperature, soil pH, and rhizosphere microorganisms in different batch experiments [21,22]. Hairy roots, which possess all the morphological and physiological characteristics of normal roots and allow indefinite propagation, have been proven as a convenient experimental tool for investigating the interactions between plant cells and metal ions [22,23,24,25]. The hairy roots of *Thlaspi caerulescens* were chosen to study oxidative stress and the response of the antioxidant defense system under Cd stress [26]. The hairy roots of *Trifolium repens* was used as a sensitive tool for monitoring and assessing Cd contamination in the environment [27]. Notably, further genetic modification via the Ri plasmid of *Agrobacterium rhizogenes* is quite straightforward for the introduction of enhanced Cd accumulation traits [23]. Given these attractive characteristics of the hairy root culture process, it has been chosen as a model plant system to investigate the Cd tolerance of hyperaccumulating plants in this work.

Studies on the Cd tolerance mechanisms of hyperaccumulating plants are of great importance [28]. It was reported that glutathione (GSH) is an important non-enzymatic antioxidant in cells which can effectively scavenge free radicals produced by Cd stress and can reduce peroxidative damage [29]. During this process, GSH synthetase (GSHB), glutathione *S*-transferase (GST) and glutathione reductase (GSR) also play an important role in the intracellular detoxification of intracellular xenobiotics and toxic compounds [30,31,32]. Three basic helix-loop-helix transcription factors (FER-like Deficiency Induced Transcripition Factor (*FIT*), *AtbHLH38*, and *AtbHLH39*) of *Arabidopsis* were reported to be involved in the plant’s response to Cd stress, and the tolerance of transgenic *Arabidopsis* to Cd was significantly improved [33]. Moreover, an RNA-Seq method, which can be used to analyze gene expression, has been successfully developed to further investigate the molecular mechanism of response to various biotic and abiotic stresses in various plants [8,24,34,35,36,37]. Data on transcriptome differences indicate that the difference in the iron-deficient transporter gene in response to Cd stress was found to be related to Cd uptake and redistribution in both *Solanum nigrum* L. and *Solanum torvum* L. [38]. Another study of comparative transcriptomics showed that the transporter gene NcNramp1 had higher expression in the roots and shoots of the elite Cd accumulator Ganges in response to Cd stress [39]. The gene NcNramp1, which is one of the major transporters involved in Cd hyperaccumulation in *Noccaea caerulescens*, appears to be the main cause of high Cd accumulation by increasing *Noccaea caerulescens* gene expression [39].

Notably, among different candidates for phytoremediation of Cd pollution, *Brassica* species are considered promising hyperaccumulators [40,41]. A previous study reported that most of the metal transporter genes (74.8%, 202/270) responded to Cd stress, suggesting that at least some of them are involved in Cd uptake and translocation in *Brassica napus* [42]. miR158 may be required for *Brassica napus* tolerance to Cd by decreasing BnRH24 levels [43]. A total of 84 ATP binding cassette (ABC) genes of *Brassica napus* were up-regulated under Cd stress [44]. The response mechanism of the hairy roots of the *Brassica* species to Cd stress is of particular significance in the development of an efficient phytoremediation process based on hairy root cultures, genetic modification, and subsequent regeneration of the whole plant. However, studies on the absorption, accumulation, and tolerance of Cd in the hairy roots of *Brassica campestris* L. are still limited, and the molecular mechanism remains unclear.

The hairy roots of *Brassica campestris* L. could survive in a medium with the addition of 100–200 μM CdCl_2_ [45,46], as seen in our previous studies. The high Cd accumulation in the hairy roots exceeded 1000 mg/kg [47], which is significantly higher than the threshold level of 100 mg Cd/kg for a Cd-hyperaccumulating plant [46] and comparable to those (204–7408 mg/kg) obtained in whole Cd-hyperaccumulating plants [48,49]. Established hairy roots of *Brassica campestris* L. have been chosen as a model system for our further studies on the plant’s response under Cd stress. Transcriptome sequencing by RNA-Seq is performed to investigate the gene transcription pattern and the related molecular mechanism of hairy roots exposed to Cd. Moreover, the function of some distinguished genes such as GSHB and GST also are verified, providing important information for future genetic modification of hairy roots and regeneration of the whole plant with enhanced abilities.

## 2. Results

### 2.1. An Overview of the mRNA of Brassica campestris L. under Cadmium Stress

To investigate the response of the Cd hyperaccumulator *Brassica campestris* L. to Cd stress, we analyzed and compared mRNA expression profiles of Cd-treated and untreated hairy roots of *Brassica campestris* L. Six samples were measured using high-throughput sequencing, and the average data volume of one sample was 6.72 GB (Table 1). Clean reads were obtained from raw reads by removing reads with poor contaminants, low mass, or a high N content of unknown bases. The Q20 value of all clean reads of samples exceeded 98%, and the error rate was less than 1% (Table 1), indicating that the quality of the sequencing met the requirements of subsequent analysis. There were no AT or GC separations in the sequencing report, indicating that sequencing was stable (Appendix A). Overall, if the proportion of low quality (quality < 20) bases was low, the sequencing quality was good (Appendix A). After obtaining clean reads, the clean reads were aligned to the reference genome sequence by Hierarchical Indexing for Spliced Alignment of Transcripts (HISAT). The average ratio of each sample reached 64.97% (Appendix A), and the uniform ratio between samples indicated that the clean reads data between samples were comparable. To conclude, the transcriptome sequencing results were credible.

According to the untreated and treated samples, the similarity heat map shows that the correlations among control replicates C1, C2, and C3 were relatively high; the correlations among replicates within the Cd treatment T1, T2, and T3 were comparatively high. This reveals that Cd treatment caused a large difference in the transcriptome of hairy roots of *Brassica campestris* L. (Figure 1A). Through the framed genetic tree (Figure 1B), it is further illustrated that the closer the expression profile, the higher the correlation, which indicates that Cd stress greatly changed the expression profile of the hairy roots of *Brassica campestris* L. Above all, the results of transcriptome sequencing were reliable. 

### 2.2. Analysis of Differentially Expressed Genes (DEGs)

A total of 35,385 genes were detected by sequencing C and T samples. Among these genes, 30,873 genes were found in both C and T samples, while 2490 and 2022 genes were observed in C and T, respectively (Appendix A). There was a significant difference in unigenes between T and C (*p* < 0.05), of which 1564 unigenes were up-regulated and 830 were down-regulated after CdCl_2_ treatment (Figure 2.).

### 2.3. Gene Ontology (GO) Functional Analysis of DEGs

Following the identification of DEGs, we analyzed their functions using GO; we classified and enriched their GO function. Detailed proportions of the GO annotation for DEGs are shown in Figure 3, indicating that molecular functions, biological processes, and cellular components were well represented. The GO functional classification results show that the DEGs were mainly enriched in 47 GO terms (Figure 3). It was found that the biological functions of the DEGs in the hairy roots of *Brassica campestris* L. were enriched in cell stimulation responding to Cd stress. Among the most abundant genes in hairy roots were those enriched in response to stimulus (43.6%), an organic substance (15.9%), an endogenous stimulus (14.0%), a chemical (21.2%), a carbohydrate (4.6%), stress (21.2%), oxygen-containing (6.4%) and hormone (11.8%) (Appendix A).

### 2.4. Pathway Functional Analysis Results

Different genes in the hairy roots coordinate with each other to perform their biological functions, and pathway-based analysis helps to further understand the biological functions of genes. We mapped the DEGs to the reference canonical pathways in the Kyoto Encyclopedia of Genes and Genomes (KEGG) database to further identify the active metabolic pathways involved in the responses to Cd. The pathway classification results are shown in Figure 4. First, we divided the differentially expressed genes into the following six categories: metabolism, organic systems, environmental information processing, genetic information processing, cellular processes, and human diseases. Then, we further divided the six categories into 20 subcategories. We listed the pathway and the top six differential gene enrichments, as shown in Appendix A. It can be seen from the table that Cd stress caused changes in the hairy root metabolic pathway, secondary metabolites, plant–pathogen interactions, plant hormone signal transduction, starch and carbohydrate metabolism, and RNA transport in *Brassica campestris* L. Altogether, there were 433 significant differentially expressed genes involved in the metabolic signaling pathway, accounting for 24.41% of the total number of differentially expressed genes in the signaling pathway (Appendix A).

### 2.5. Real-Time PCR Confirmation of the Gene Expression 

To verify the reliability of sequencing, eight differentially expressed genes were randomly selected for RT-PCR validation. Real-time PCR results showed that the transcription levels of these eight genes were consistent with the results of transcriptome sequencing (Figure 5, Table 2). Using this, we can conclude that the results of transcriptome sequencing can represent differences in transcriptome levels in the hairy roots of *Brassica campestris* L. caused by Cd stress. We found that the Gene ID 103874543 involved in the synthesis of glutathione was up-regulated.

### 2.6. Identification of Key Gene/Protein and Expression Validation 

The GO and KEGG analysis of transcriptome sequencing showed that the GSH metabolic pathway plays an important role in regulating the Cd enrichment ability and defense ability of rapeseed hairy roots. Therefore, we continue to study the GSH metabolic pathway. Using the KEGG database, we obtained the following signal path map (Appendix A). By analyzing key genes in the GSH metabolic signaling pathway, significant up-regulation of GSHB and GST in transcriptome sequencing results caught our attention, so we selected validation of GSHB and GST protein expression levels.

Western Blot indicated that the expression of GSHB at 200 μM Cd stress was 1.85-fold higher than that of the control group, and the expression level of GST was 2.57-fold higher than that of the control group. The amount of actin protein was stable across the treatments (Figure 6). This result further demonstrated that, not only the mRNA, but also the proteins of GSHB and GST were highly upregulated in the hairy roots of *Brassica campestris* L.

## 3. Discussion

It is particularly important to investigate the Cd tolerance and the related molecular mechanisms of the hairy roots of *Brassica campestris* L. [47,50,51]. Our previous studies showed that the high Cd accumulation in the hairy roots of *Brassica campestris* L. reached more than 1000 mg/kg [47]. The growth of hairy roots of *Brassica campestris* L. was significantly inhibited when it grow in the medium with CdCl_2_ of 200 μM for 24 h [47]. Under this condition, the hairy roots indicated obvious browning and decaying. Additionally, a high Malondialdehyde (MDA) content and microscopic observation of the hairy roots indicated that cell membrane damage and apoptosis appeared, which increased our interest to further investigate the response molecular mechanism using RNA-Seq to perform transcriptome sequencing.

Results indicate that the hairy roots of *Brassica campestris* L. may improve their Cd tolerance and response to the Cd stress as follows. First, the synthesis of some antioxidants (e.g., GSH and AsA) in the hairy roots may be enhanced by up-regulating expression of the related genes (Figure 7). The KEEG analysis and Western Blot test indicated that the GSHB genes were significantly up-regulated and the GSHB content was improved, hence definitely benefiting the corresponding GSH synthesis (Figure 6 and Appendix A). When considering GSH as one of the main antioxidants in plant cells [52,53], it, therefore, is expected to enhance the scavenging of reactive oxygen species (ROS) [54].

Second, detoxification of the hairy roots from Cd stress could be further enhanced by up-regulating the expression of the genes involved in other GSH metabolism aside from synthesis catalyzed by GSHB (Figure 7). The reaction of scavenging ROS can be catalyzed by the enzymes including GPX or GST with GSH as the substrate. Described in the Results section, the GST and GPX (103830386, 103862090) genes in the glutathione metabolic pathway were up-regulated and the enhanced GST synthesis also was verified by a Western Blot test (Figure 6 and Appendix A). Conversely, the up-regulated G6PD (103836988) gene in the Cd-treated hairy roots (Appendix A) benefitted the synthesis of G6PD, which catalyzed glucose 6-phosphate to produce a co-enzyme (i.e., NADPH) of GSR and help to regenerate GSH from GSSG [55]. Previous studies also observed that the expression of genes related to some key enzymes of GSH metabolism, including ROS scavenging (e.g., GPX and GST) and GSH regeneration from GSSG (e.g., GSR), increased in response to different heavy metal stress (e.g., Ni, Cd etc.) [55,56,57].

Third, the hairy roots of *Brassica campestris* L. may improve their tolerance to Cd by metal-binding and/or metal-chelating reactions by GSH and/or phytochelatins (PCs). As a major non-enzymatic antioxidant, GSH also may be involved in cell defense against toxicants via binding them or their metabolites with catalysis of GST [53,58]. Previous studies have shown that high levels of expression of most GST genes result in higher GST activity, which increases maize root tolerance to abiotic stress [59]. Data from the GO analysis, regarding the GST enzyme gene in this work, indicate that GST mainly plays a role in the toxin metabolic process and the response to metal ions (Appendix A). Alternately, PCs may play an important role in the detoxification of heavy metals in this process [60]. Data analyses indicate that the expression of the glutamate-cysteine ligase catalytic subunit (103861360), which participates in cysteine and methionine metabolism following GSH synthesis and synthesis of subsequent chelating agents (e.g., PCs) with GSH as a precursor, were up-regulated in plants under Cd stress (Figure 7). Previous studies also found that a large number of PCs were observed in rice plants growing under heavy metal stress [53,61]. The functional *Vicia sativa* Phytochelatin Synthase 1 (PCS1) homolog is overexpressed in *Arabidopsis thaliana* [62]. The expression levels of GSHB, GSR, PCs, and other chelating peptide synthesis-related enzymes in hyperaccumulators under heavy metal stress were significantly increased [63,64,65,66,67].

To summarize, all these factors could provide an integrated protection for hairy roots from Cd stress. Together with previous studies, results in this work present a better understanding of the regulation network of the hairy roots of *Brassica campestris* L. under Cd stress, hence providing important information for further genetic modification and subsequent regeneration of the whole plant with improved accumulation abilities.

## 4. Materials and Methods 

### 4.1. Plant Material L and Cadmium Stress Treatment

The hairy roots of *Brassica campestris* L. used in this study were previously induced by infection of the seedlings with *Agrobacterium rhizogenes* in our laboratory [45]. The hairy roots were maintained in Murashige and Skoog medium (with vitamins and sugar) medium (Coolaber, Beijing, China) based on previous methods [45]. Briefly, 0.5 g of fresh hairy roots of *Brassica campestris* L. were placed in a 100 mL pyridoxine flask containing 50 mL Murashige and Skoog medium (with vitamins and sugar) liquid medium (NH_4_NO_3_ 1650 mg/L, CaCl_2_ 332.2 mg/L, MgSO_4_ 180.7 mg/L, KNO_3_ 1900 mg/L, KH_2_PO_4_ 170 mg/L, H_3_BO_3_ 6.2 mg/L, CoCl_2_·6H_2_O 0.025 mg/L, CuSO_4_·5H_2_O 0.025 mg/L, FeNaEDTA 36.7 mg/L, MnSO_4_·H_2_O 16.9 mg/L, Na_2_MoO_4_·2H_2_O 0.25 mg/L, KI 0.83 mg/L, ZnSO_4_·7H_2_O 8.6 mg/L, glycine 2 mg/L, moy-inositol 100 mg/L, nicotinic acid 0.5 mg/L, pyridoxine HCl 0.5 mg/L, thiamine HCl 0.1 mg/L) and incubated at 100 r/min and 28 °C for 5 d. Our previous results showed that the growth of hairy roots of *Brassica campestris* L. was significantly inhibited when grown in a medium with 200 μM CdCl_2_ for 24 h [47]. Therefore, this treatment condition was chosen for further response mechanism study in this work. The Cd-treated hairy roots of *Brassica campestris* L. were collected and rinsed three times with deionized water. The samples were then dried and stored at −80 °C. Untreated and Cd-treated samples were labeled C and T, respectively. There were three replicates for each treatment.

### 4.2. Construction and Sequencing of mRNA Libraries

Total RNA was extracted using a TRIzol reagent and was treated with DNase to remove genomic DNA contamination. The eukaryotic mRNA was enriched with Oligo(dT) magnetic beads, the interrupter reagent was used to break the mRNA into short fragments in a Thermomixer (Eppendorf, Hamburg, Germany) to synthesize a strand of cDNA using the interrupted mRNA as a template and, then, the preparation was performed. A two-strand synthesis reaction system synthesized a double-strand cDNA and then used a kit for purification and recovery, cohesive end repair, the addition of base “A” to the 3’ end of the cDNA, and the attachment of a linker. Subsequently, the fragment size was chosen, followed by PCR amplification. A suitable library was inspected with an Agilent 2100 Bioanalyzer and an ABI StepOne Plus Real-Time PCR System. Then, we performed paired-end sequencing on an Illumina Hiseq4000 (BGI, Shenzhen, China) following the manufacturer’s protocol. The six RNA libraries consisted of three control libraries and three Cd-treated libraries.

### 4.3. Sequence and Primary Analysis

We used the Illumina paired-end RNA-seq approach to sequence the hairy roots of *Brassica campestris* L. transcriptomes, each producing 6 Gb of multiple data, resulting in a total of 40.32 Gb of sequences. Prior to assembly, low-quality readings were removed, including reads containing sequencing adaptors, sequencing primers, and nucleotides with a quality score below 20. The original sequence data were submitted to the NCBI Database under registration number PRJNA543954.

### 4.4. RNA-Seq Reads Mapping

We compared the readings of the different samples to the *Brassica campestris* L. (Field mustard) ID: 229 cabbage canola reference genomic sequence using the HISAT2 software. The alignment process can be divided into the following three parts: (1) aligning of reads to a known transcriptome (optional); (2) aligning of the aligned pairs of reads on the reference genome; (3) unpaired read segments are aligned to the reference genome. Reads aligned to the specified reference genome were called mapped reads, and subsequent information analysis was performed based on mapped reads.

### 4.5. Transcript Abundance Estimation and Differential Expression Testing

The results of the HISAT2 [68] alignment to the reference genome were submitted to htseq-count (v0.6.0) [69] for processing, and the read count of each transcript was obtained. Single gene expression levels were calculated using readings per kilobase read (RPKM), which eliminated the effects of gene length and sequencing levels during the calculation of gene expression, making the samples comparable. Then, we used DEGseq for analysis, using the normalization method of quantiles, fold change ≥2, and *p*-value < 0.05 as the threshold for determining whether the gene was differentially expressed to obtain DEGs.

### 4.6. Gene Annotation, Classification, and Metabolic Pathway Analysis

To study the functional partitioning of DEGs in the hairy roots of *Brassica campestris* L. under Cd stress, we used GO and KEGG for further annotation, classification, and metabolic pathway analysis [70,71,72]. First, a gene ontology (GO) enrichment analysis of differentially expressed genes was performed by the clusterProfiler R package in which the gene length deviation was corrected. A GO term with a corrected *p*-value of less than 0.05 was considered to be significantly enriched by differentially expressed genes. The KEGG pathway was retrieved from the KEGG web server (http://www.genome.jp/kegg/). The clusterProfiler R package was used to test the statistical enrichment analysis of differentially expressed genes in the KEGG pathway.

### 4.7. Quantitative Real-Time PCR Validation 

Eight DEGs were randomly selected for RT-PCR validation. The primer sequences and reference genes of these genes are listed in Appendix A. Total RNA (0.2 μg) from each root sample was reverse transcribed using a PrimScript^®^ RT Kit (Takara, Beijing, China) and random primers according to the manufacturer’s instructions. Quantitative PCR reactions were performed in a 20 µL reaction volume using a Promega GoTaq^®^ qPCR Master Mix Real Time PCR kit (Takara, Beijing, China) according to the manufacturer’s instructions. The reaction was carried out on anSLAN-90P (Hongshi Medical Technology Co., LTD., Shanghai, China). Each biological replica was technically replicated three times. Relative RNA expression of the selected genes, which is the expression of these genes relative to an internal reference gene, was used as an indicator of the genes’ expression in different samples [73,74,75]. The relative expression levels of the selected genes were calculated using the 2-ΔΔCT method and the probable ubiquitin-conjugating enzyme E2 21 (BnUBC21) gene was used as a reference gene to correct gene expression [76]. Three replicates were performed for each sample.

### 4.8. Western Blot Analysis

The extraction of total hairy root protein was carried out using a Plant Protein Extraction Kit (CoWin Biosciences, Beijing, China). The extracted proteins were assayed by a Pierce™ BCA Protein Assay Kit (Thermo Fisher Scientific, Massachusetts MA, USA). Standard Western blots were performed. Immunoblotting was carried out by using the following primary antibodies: GSH-S (Agrisera AB, Vännäs, Swedish), GST class-phi (Agrisera AB, Vännäs, Swedish), and β-actin (CoWin Biosciences, Beijing, China). After incubation with the secondary antibodies, the signal was developed by chemiluminescence and autoradiography. Densitometric analysis was performed using ImageJ software (National Institutes of Health, Bethesda, NY, USA).

### 4.9. Statistical Analysis

All the experimental data were obtained with three or more repetitions. The data obtained from the experiment were analyzed using IBM’s SPSS 20 software by one-way analysis of variance (ANOVA). Statistical analysis was calculated by Duncan’s method (*p <* 0.05).

## 5. Conclusions

During this study, 2364 DEGs were discovered under Cd stress in the hairy roots of *Brassica campestris* L. based on RNA-Seq analysis. These genes were mainly involved in transcription-related processes, defense, stress responses, and transport processes in the response of *Brassica campestris* L. to Cd stress. Furthermore, data from Western blot tests indicated that the signaling pathway for glutathione synthesis and metabolism played an important role in the response to Cd stress. These results provide valuable information for enhancing our understanding of the heavy metal tolerance of *Brassica campestris* L. and the corresponding molecular mechanism. It is expected that the Cd-accumulating ability will be further improved by combining transgenic modification of the hairy roots, such as over-expression of the genes involved in Cd hyperaccumulation with subsequent plant regeneration, hence making the whole regenerated plant of *Brassica campestris* L. more promising for future application in phytoremediation.

## Figures and Tables

**Figure 1 ijms-21-00180-f001:**
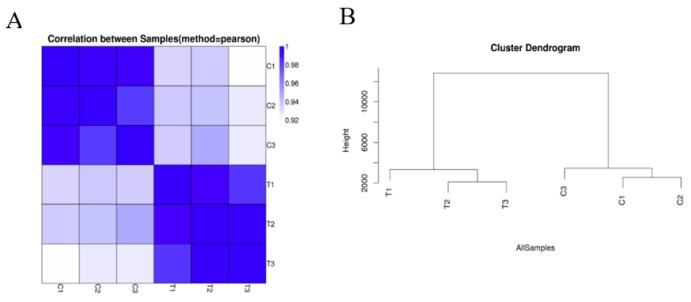
Sample quality test. (**A**) Heatmap of Pearson correlation between samples. The x- and y-axes represent each sample. (**B**) Hierarchical clustering between all and samples. C1, untreated hairy roots group number 1; C2, untreated hairy roots group number 2; C3, untreated hairy roots group number 3; T1, hairy roots treated with 200 μM Cd for 1 d, group number 1; T2, hairy roots treated with 200 μM Cd for 1 d, group number 2; T3, hairy roots treated with 200 μM Cd for 1 d, group number 3.

**Figure 2 ijms-21-00180-f002:**
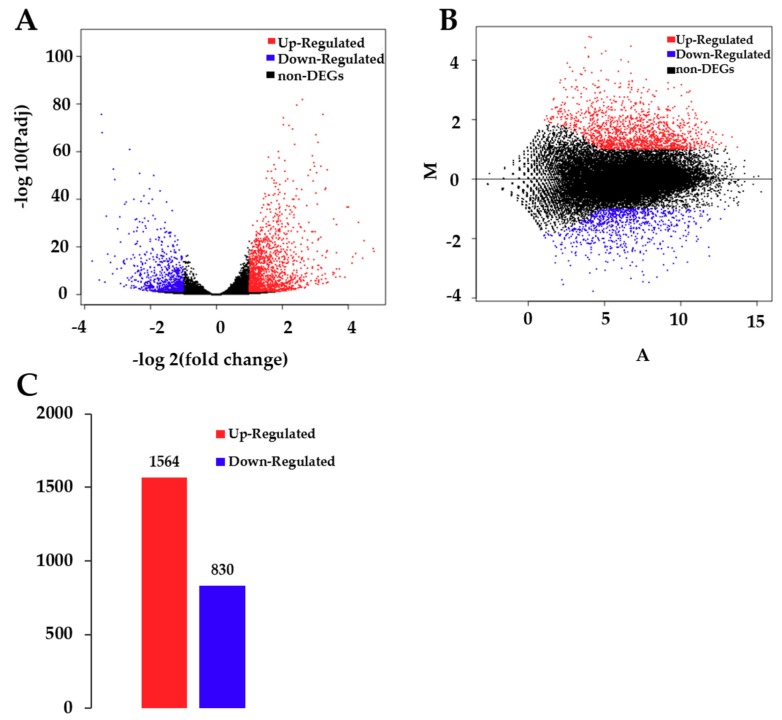
Analysis of differentially expressed genes (DEGs) between control and treatment groups. (**A**) Volcano plot of DEGs. The x-axis represents the log2-transformed fold change. The y-axis represents the -log10-transformed significance. (**B**) Minus-versus-add (MA) plot of DEGs. The x-axis represents value A (log2-transformed mean expression level). The y-axis represents value M (log2-transformed fold change). Red points represent up regulated DEGs. Blue points represent down regulated DEGs. Black points represent non-DEGs. (**C**) Summary of DEGs. The x-axis represents the compared samples. The y-axis represents DEG numbers. Red color represents up-regulated DEGs. Blue color represents down-regulated DEGs.

**Figure 3 ijms-21-00180-f003:**
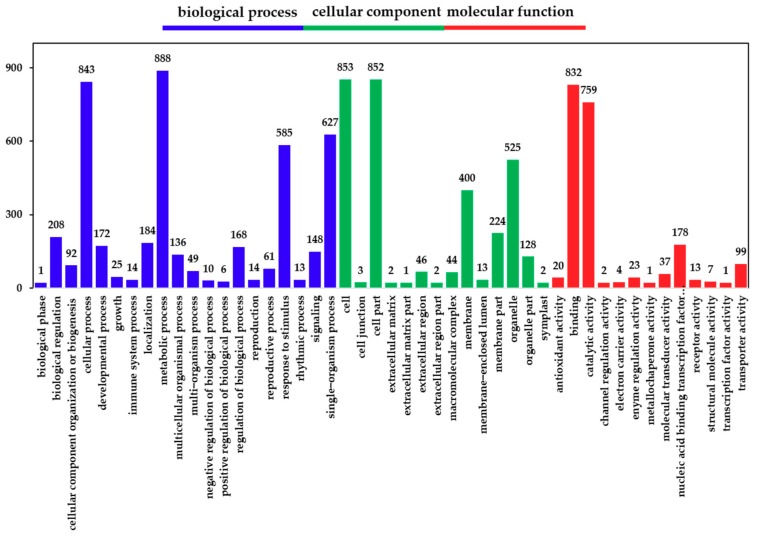
Gene ontology (GO) classification of DEGs. The x-axis represents the GO term. The y-axis represents the number of DEGs.

**Figure 4 ijms-21-00180-f004:**
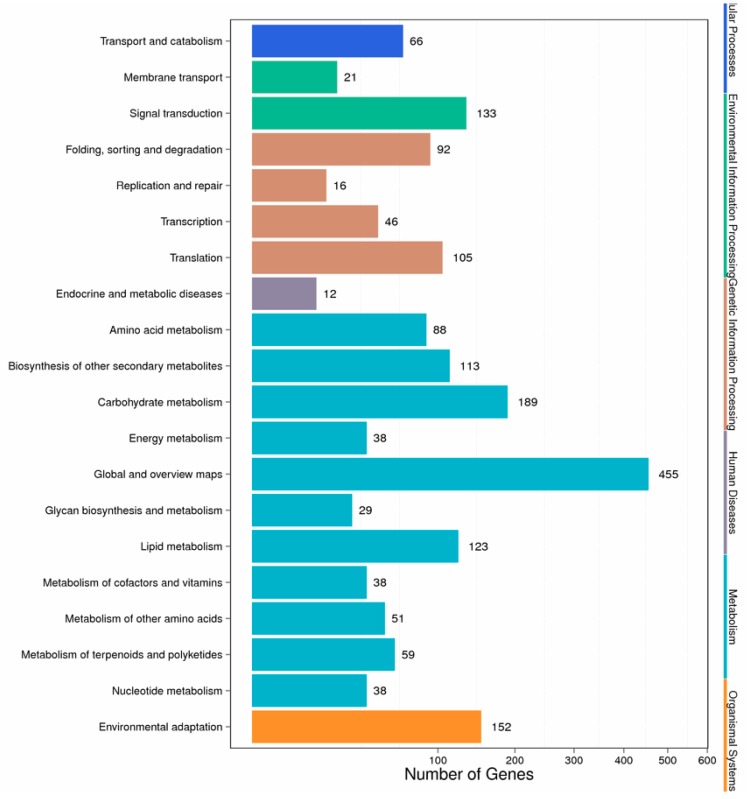
Pathway classification of DEGs. The x-axis represents the number of DEGs. The y-axis represents the pathway name.

**Figure 5 ijms-21-00180-f005:**
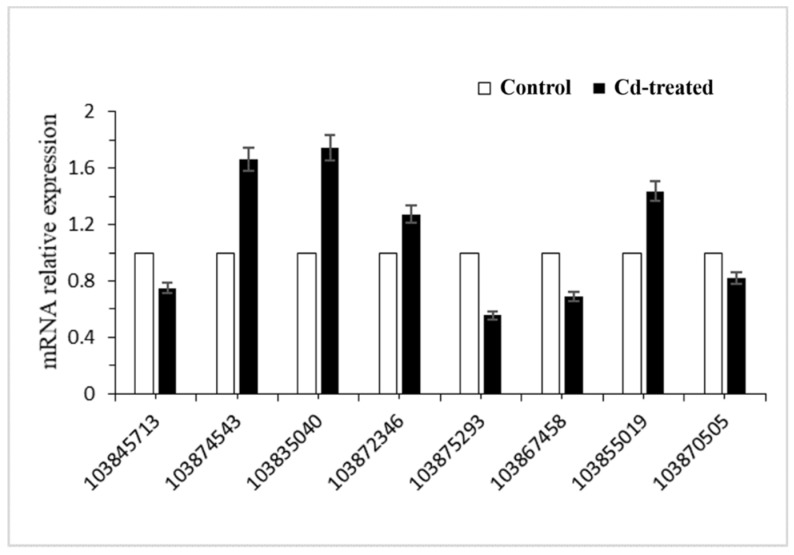
Relative expression of significant differentially expressed genes in hairy root cultures of *Brassica campestris* L.

**Figure 6 ijms-21-00180-f006:**
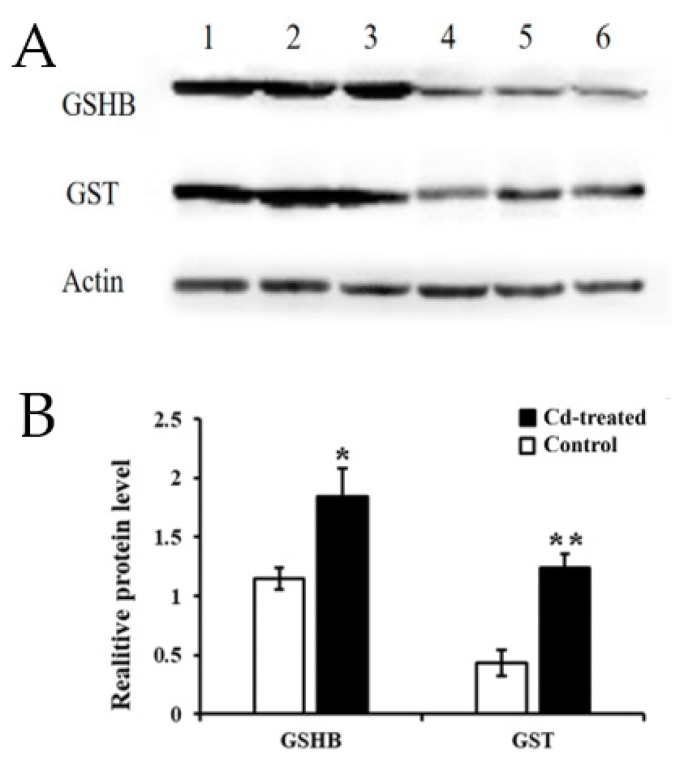
Relative expression of target genes in the hairy roots of *Brassica campestris* L. (**A**) Expression of GST and GSH protein. Lines 1, 2, 3 Cd treatment (200 µM for 24 h); 4, 5, 6: control group; (**B**) Grayscale analysis. (* *p* < 0.05, ** *p* < 0.01).

**Figure 7 ijms-21-00180-f007:**
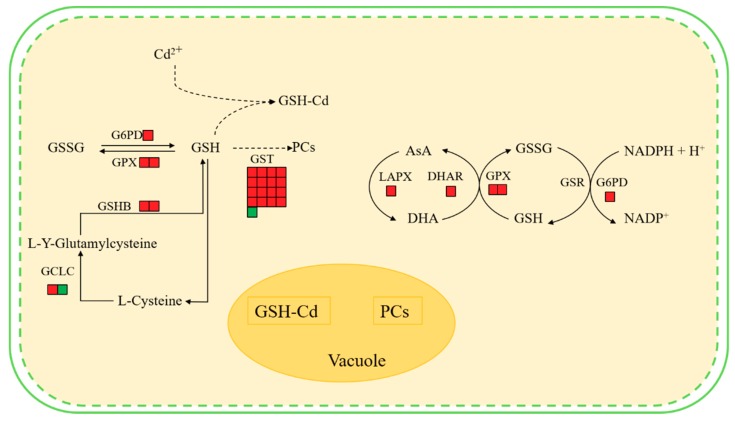
Main changes in *Brassica campestris* L. metabolism after Cd exposure. Red boxes indicate an increased abundance in Cd-treated plants compared with controls, and green boxes indicate a reduced abundance in Cd-treated plants compared with controls. Abbreviations: GSHB, glutamine synthase; GSH, glutathione; GSSG, oxidized glutathione; GST, glutathione-*S*-transferase; GPX, glutathione peroxidase; GCLC, glutamate-cysteine ligase catalytic subunit; GSH-Cd, glutathione-cadmium complex; PCs, phytochelatins; AsA, ascorbate; MDH, malate dehydrogenase; MDHA, monodehydroascorbate; LAPX, l-ascorbate peroxidase; G6PD, glucose-6-phosphate 1-dehydrogenase; GSR, glutathione reductase.

**Table 1 ijms-21-00180-t001:** Summary of sequence reads for six RNA samples including three replicate control treatments (C1–C3) and three samples of cadmium treatment (T1–T3).

Sample	Raw Reads (Mb)	Clean Reads (Mb)	Clean Bases (Gb)	Clean Reads Q20 (%)	Clean Reads Q30 (%)	Clean Reads Rate (%)
C 1	53.34	44.28	6.64	98.26	94.45	83.01
C 2	54.83	45.37	6.81	98.37	94.75	82.74
C 3	56.47	45.28	6.79	98.39	94.82	80.19
T 1	54.85	45.03	6.75	98.34	94.67	82.09
T 2	54.77	44.14	6.62	98.34	94.69	80.6
T 3	54.88	44.71	6.71	98.28	94.52	81.48

**Table 2 ijms-21-00180-t002:** Differential expression of eight target genes (relative to reference genes) quantified by qPCR in control and Cd-treated hairy root samples.

Gene ID	*Nr* Description	C-Expression	T-Expression	log2FoldChange (T/C)	*p-*Value
103845713	peroxidase N	1547.49	311.39	–2.31	4.20 × 10^−10^
103874543	glutathione synthetase	406.26	1117.96	1.46	1.86 × 10^−16^
103835040	calcium-transporting ATPase 2	402.56	974.08	1.27	3.25 × 10^−16^
103872346	ABC transporter G family member 40	532.77	4004.34	2.91	3.07 × 10^−62^
103875293	aquaporin TIP1-2	2508.98	1039.61	–1.27	3.03 × 10^−18^
103867458	aquaporin NIP2-1	2308.78	703.43	–1.71	5.19 × 10^−47^
103855019	protein NRT1/ PTR FAMILY 2.11	584.22	2262.11	1.95	3.82 × 10^−34^
103870505	ras-related protein RABC2b	16.88	1.44	–3.55	4.90 × 10^−8^

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
