# Peer review of "Comparative Transcriptome Analysis of the Molecular Mechanism of the Hairy Roots of Brassica campestris L. in Response to Cadmium Stress"

_ijms, 2019, doi:10.3390/ijms21010180_

Round 1

Reviewer 1 Report

The authors of the manuscript "Comparative Transcriptome Analysis of Molecular Mechanism of the Hairy Root of Brasssica campestris L. in Response to Cadmium Stress" present interesting results of identification of Cd stress-responsive genes. The manuscript makes a significant contribution to the understanding of complicated plant responses to Cd stress. The work appears novel and is within the scope of the journal. 

There are some remarks.

1. After the first mentioning of cadmium (Cd) there should be Cd used constantly. The same comment with regard to reactive oxygen species there should be used ROS.

2. The Brasssica campestris L. is written as Brasssica campestris L. and Brasssica campestris L.. All Latin names of the species should be checked.

3. There is no A, B and C designation in the Figure 2. Line 140.

4. The description of the Figure 5 and 6. should be changed. I think it is better to use e.g. "Cd-treated" instead of "Add Cadmium".

5. Table 2 description: "Signafication of differently expressed genes" should be corrected. Also "Pvalue" should be changed to "P-value".

6. There are sometimes CdClbut also CdCl2 (line 135, 138, 242) and H2O2 (line 286).

7. There should be "phytochelatins" not "Phytochelins" - line 318.  

8. The sentences: lines 244-247, lines 268-269 should be corrected.

9. The authors sometimes write Fig. (e.g. line 312, 313) and sometimes Figure (e.g. line 339, 341, 346, 348) in the discussion. 

10. There is also different front used in the text: lines 256-262, 421-422.  

Author Response

Point 1: After the first mentioning of cadmium (Cd) there should be Cd used constantly. The same comment with regard to reactive oxygen species there should be used ROS.

Response 1: Thank you for your comments. We have checked the full text carefully. We used Cd (lines 27,29,166,183,184,187 and 222) after the first mention of cadmium and used ROS instead of reactive oxygen species (line 301, line 308).

Point 2: The Brasssica campestris L. is written as Brasssica campestris L. and Brasssica campestris L.. All Latin names of the species should be checked.

Response 2: We are very sorry for our negligence. We have revised all Latin names to a standard format: “Brassica campestris L.” (line 187, lines 221–222, line 279)

Point 3: There is no A, B and C designation in the Figure 2. Line 140.

Response 3: Thank you for your comments. The letters (A, B and C) were combined with pictures in Figure 2 (line 205).

Point 4: The description of the Figure 5 and 6. should be changed. I think it is better to use e.g. "Cd-treated" instead of "Add Cadmium".

Response 4: According to your kind comment, we have used "Cd-treated" instead of "Add Cadmium" in Figures 5 and 6 (line 260, line 278).

Point 5: Table 2 description: "Signafication of differently expressed genes" should be corrected. Also "Pvalue" should be changed to "P-value".

Response 5: We have changed the description "Signafication of differently expressed genes" to “Significant differently expressed genes” by referring to relevant literature (line 263). Also, "Pvalue" has been changed to "p-value" (line 263, line 539).

Point 6: There are sometimes CdCl2 but also CdCl2 (line 135, 138, 242) and H2O2 (line 286).

Response 6: Thank you for your comments. All cases of “CdCl2” have been corrected to “CdCl2 (lines 204)”, and H2O2 was deleted due to the modification of the Discussion part of the article. Based on the Reviewer's suggestion, we paid attention to the correct writing of all subscripts.

Point 7: There should be "phytochelatins" not "Phytochelins" - line 318.

Response 7: Thank you for your comments. This error has been corrected in the revised MS (lines 226–227).

Point 8: The sentences: lines 244-247, lines 268-269 should be corrected.

Response 8: Thank you for your comments. The description of the sentences was rewritten in the revised MS (lines 399–400).

Point 9: The authors sometimes write Fig. (e.g. line 312, 313) and sometimes Figure (e.g. line 339, 341, 346, 348) in the discussion.

Response 9: We have used Figure uniformly in the revised MS (lines 175–176, lines 104–107, line 185, line 275).

Point 10: There is also different front used in the text: lines 256-262, 421-422.

Response 10: We have made corrections according to the Reviewer’s comment (lines 292–298, lines 569–579).

Other changes in the MS

All changes in the revised manuscript have been marked in red. According to Suggestions of Reviewers/Editor, we used MPDI English editing service for our manuscript. Another grant number, 2017JBM073, was added due to support for the Article Processing Charges (APC) of this MS (Lines 593–596). The Introduction, Discussion, and Conclusions sections have been revised, and the literature references have been numerically re-listed in the Reference section. Due to the many changes in the article, we supplement the abbreviations (Line 599).

Reviewer 2 Report

This study shows that hairy roots of Brassica campestris show a considerable number of up and down regulated genes when exposed to 200 µM of cadmium in liquid medium over 24 h. Analysis of gene functions indicates regulation of sulphur salvage pathways, including metabolism of glutathione and phytochelatin. This is a useful conclusion, although not quite novel.

The paper suffers from a lack of focus. The choice of Brassica campestris is motivated by its possible use in phytoremediation, but the observations do not contribute to a better phytoremediation at all. How the possible use of this plant species in phytoremediation is promoted by the present results is not discussed. The authors mention that the plants used were genetically modified but it is unclear what was the modification and whether this could interfere with the present results.

Another aspect of study seems to be the mechanism of tolerance to cadmium stress, however, no data are provided on the tolerance itself (e.g. by comparison among different populations of B. campestris). Finally the focus on hairy roots as a target tissue for gene expression makes the work difficult to interpret, since toxicity of cadmium is not limited to hairy roots and an important aspect of cadmium and zinc tolerance is translocation of the metal to other tissues. From studies on hairy root tissue alone one cannot obtain a coherent picture of responses to cadmium.

It seems that the effects of cadmium on the phenotypic level have been presented in a parallel paper. The present data are better merged with that paper.

There is too much literature review in the Introduction, too much Method description in the Results section, and too much recapitulation of Results in the Discussion.

Editorial remarks, illustrating imprecise expression and unclarities:

Line 104: cadmium hyperaccumulator – what do you mean by that?

Line 109: Q20 – please explain

Line 110: better – better than what?

Line 110: AT or GC separations – what do you mean by that?

Line 112: >20 – what does that mean?

Line 114: data – what data?

Line 119: correlations – correlations with what?

Line 120: The average ratio – ratio for what?

Line 136: The total number of genes expressed was 30873, but there were 2490 in the control group and 2022 in the treated plants – how come the total number is so much larger than the numbers in either group?

Line 141: Differentially expressed – you mean differential between control and trne eatment groups?

Line 150: GO = Gene Ontology

Line 191: What are the units for “relative RNA expression? (relative to what?). “Contrast” = Control and “Add Cadmium” = Treatment?

Line 192: transgenic?

Lines 213- 221: This is all general knowledge, not fit for the Discussion section.

Line 231: MDA experiments - what kind of experiments are that?

Line 277: super enriched – enriched with what? Do you mean hyperaccumulating?

Line 289: toxicity of Barley – is barley a toxic plant?

Line 318: phytochelins = phytochelatin

Line 331: VsPCS1: You mean Phytochelatin Synthase 1 from Vicia sativa? (green arrow pea is Pisum sativum).

Line 332: transferred – I guess you mean transduced?

Line 335 – 350: This is all recapitulation of results, you shouldn’t do that in the Discussion.

Line 353: transgenic?

Line 356: Details are needed about the type of liquid medium and its composition.

Line 430-437: These are not real conclusions, but recapitulations of your results.

Author Response

Dear Reviewers:

Thank you very much for your letter and for editing the manuscript (MS) entitled “Comparative Transcriptome Analysis of the Molecular Mechanism of the Hairy Root of Brassica campestris L. in Response to Cadmium Stress” (ID: ijms-632636). We would like to thank you for the time and effort spent reviewing our paper. The comments are very helpful for improving the quality of our work. Motivated by the comments, we have carefully revised the manuscript and tried to fix all the problems you mentioned. All authors have seen the revised manuscript/response letter and approved to submit them to this excellent journal.

In brief, the literature review in the “Introduction” section was substantially shortened and polished. The “Materials and Methods” section was also revised based on the reviewers’ comments. The discussion on the recapitulation of ‘Results’ in the ‘Discussion’ section, the mechanism, and the contribution were carefully revised, supplemented, and/or enhanced. In addition, the MS has been checked and revised very carefully for typos, grammatical errors, sentence structure, and font size.

All revisions have been done using Word’s “track changes” function so that changes are easily visible. The revised MS has been uploaded for your review.

Below are our responses to the comments from all the reviewers. Thank you very much for your valuable time.

Sincerely,

All authors

Point-to-point responses

Point 1: The paper suffers from a lack of focus. The choice of Brassica campestris is motivated by its possible use in phytoremediation, but the observations do not contribute to a better phytoremediation at all. How the possible use of this plant species in phytoremediation is promoted by the present results is not discussed.

Response 1: Thank you very much for your comments. The main objective of the current study is to investigate the gene transcription pattern of hairy roots of Brassica campestris L. exposed to Cd and to illustrate the related molecular mechanism. Based on a systematic study, 2364 DEGs were discovered under Cd stress in the hairy roots of Brassica campestris L. based on RNA-Seq analysis. These genes are mainly involved in transcription-related processes, defense stress responses, and transport processes in the response of Brassica campestris L. to Cd stress. Furthermore, Western blot analysis showed that the signal pathway of glutathione synthesis and metabolism plays an important role in the response to Cd stress.

These results provide valuable information for enhancing our understanding of the heavy metal tolerance of Brassica campestris L. and the corresponding molecular mechanism. It is expected that the Cd-accumulating ability will be further improved by combining transgenic modification of the hairy roots, such as over-expressing of the genes involved in Cd hyperaccumulation, with subsequent plant regeneration, hence making the whole regenerated plant of Brassica campestris L. more promising for future application in phytoremediation.

The corresponding revisions can be seen in the Introduction, Discussion, and Conclusion sections of the revised MS (Lines 40–93, Lines 283–419 and Lines 569–579).

Point 2: The authors mention that the plants used were genetically modified but it is unclear what was the modification and whether this could interfere with the present results.

Response 2: Thank you very much for your comments. The hairy root of Brassica campestris L. induced by Agrobacterium rhizogenes is the phenotype produced by integrating the T-DNA of Agrobacterium rhizogenes into the host cell. In fact, there were no other genes transferred except for the Agrobacterium rhizogenes gene in this study.
The corresponding descriptions of “transgenic Brassica campestris L.” and “transgenic hairy root” in the MS are inappropriate and have been revised to prevent misunderstanding (Lines 471–472, Lines 261–262, Lines 242–243, Lines 509–510 and Lines 532–533).

Point 3: Another aspect of study seems to be the mechanism of tolerance to cadmium stress, however, no data are provided on the tolerance itself (e.g. by comparison among different populations of B. campestris).

Response 3: Thank you for your comments. In our previous studies, the hairy roots of Brassica campestris L. developed in our laboratory survived in the medium with the addition of 100–200 μM CdCl2 [1,2]. The Cd accumulation in the hairy roots exceeded 1000 mg/kg [3], which is significantly higher than the threshold level of 100 mg Cd/kg for a Cd-hyperaccumulating plant [2] and comparable to levels (204–7408 mg/kg) obtained in whole plants with Cd-hyperaccumulating ability [4,5]. A brief introduction about the tolerance has been added to the MS by citing those previous studies (Lines 84–89).

Liu, J.Y. Research on Agrobacterium rhizogenes mediated IRI l gene transformation in Cd  Hypemccumulator  Brassica campestris L. Beijing Jiaotong University, 2014. Liu, H.; Zhao, H.; Wu, L.; Liu, A.; Zhao, F.J.; Xu, W. Heavy metal ATPase 3 (HMA3) confers cadmium hypertolerance on the cadmium/zinc hyperaccumulator Sedum plumbizincicola. New Phytol 2017, 215, 687-698, doi:10.1111/nph.14622. Lu, Y.; Cao, Y.; Chen, Y.; Yan, Q. The physiological response and iron and potassium contents in the hairy roots of Brassica rape L. under cadmium stress. Chinese Journal of Applied & Environmental Biology 2018, 24, 8, doi:10.19675/j.cnki.1006-687x.2017.06019. Šimonová, E.; Henselová, M.; Masarovičová, E.; Kohanová, J. Comparison of tolerance of Brassica juncea and Vigna radiata to cadmium. Biologia Plantarum 2007, 51, 488-492, doi:10.1007/s10535-007-0103-z. Van Engelen, D.L.; Sharpe-Pedler, R.C.; Moorhead, K.K. Effect of chelating agents and solubility of cadmium complexes on uptake from soil by Brassica juncea. Chemosphere 2007, 68, 401-408, doi:10.1016/j.chemosphere.2007.01.015.

Point 4: Finally the focus on hairy roots as a target tissue for gene expression makes the work difficult to interpret, since toxicity of cadmium is not limited to hairy roots and an important aspect of cadmium and zinc tolerance is translocation of the metal to other tissues. From studies on hairy root tissue alone one cannot obtain a coherent picture of responses to cadmium.

Response 4: Thank you for your comments. It is true that a study on hairy roots alone cannot provide a coherent picture of responses to Cd stress when compared to the whole plant. In this work, the hairy roots of Brassica campestris L. were chosen as a model plant system to investigate the response mechanism of Brassica campestris L. to Cd stress based on the following considerations: 1) hairy roots have been proved to be a convenient experimental tool for investigating the interactions between plant cells and metal ions in many studies, 2) the ability to identify the contributions of plant cells to Cd uptake and detoxification without interference from microorganisms; and 3) further genetic modification via the Ri plasmid of Agrobacterium rhizogenes is quite straightforward for the introduction of enhanced Cd accumulation traits [6]. Hairy roots are also considered a powerful tool for screening genetic transformants prior to regeneration of whole plants with enhanced abilities. Therefore, the current work on the response mechanism of the hairy root of Brassica campestris L. to Cd stress is of particular significance in the development of an efficient phytoremediation process based on the hairy root cultures, genetic modification, and subsequent regeneration of the whole plant.

Al-Shalabi, Z.; Doran, P.M. Metal uptake and nanoparticle synthesis in hairy root cultures. Adv Biochem Eng Biotechnol 2013, 134, 135-153, doi:10.1007/10_2013_180.

Point 5: It seems that the effects of cadmium on the phenotypic level have been presented in a parallel paper. The present data are better merged with that paper.

Response 5: Yes, results about the effects of cadmium at the phenotypic level and the tolerance of the hairy roots to Cd stress have already been published in another paper, which provided important information for this study. Therefore, these data in the published paper have been cited for improving the discussion in the revised MS (Lines 283–291).

Point 6: There is too much literature review in the Introduction, too much Method description in the Results section, and too much recapitulation of Results in the Discussion.

Response 6: According to the suggestion, we have streamlined the Introduction (Lines 40–93), Results (Lines 166–168, Lines 198–202, Lines 250–254 and Lines 272–275), and Discussion (Lines 283–291) sections.

Point 7: Line 136: The total number of genes expressed was 30873, but there were 2490 in the control group and 2022 in the treated plants – how come the total number is so much larger than the numbers in either group?

Response 7: Thank you for your comments. The description about the number of genes in the MS was unclear. In this study, a total of 35,385 genes were detected from C and T samples by sequencing. Among these, 30,873 genes were found in both C and T samples, while 2490 and 2022 different genes were observed in C and T samples, respectively (Figure S3). The corresponding descriptions have been revised to prevent misunderstanding (Lines 198–202). 

Point 8: Line 141: Differentially expressed – you mean differential between control and trneeatment groups?

Response 8: The original title of Figure 2 was inappropriate and has been revised as "Analysis of DEGs between control and treatment groups" (Line 206).

Point 9: Line 150: GO = Gene Ontology

Response 9: Thank you for your comments. Both GO and Gene Ontology were used in the literature. In the revised MS, Gene Ontology was used consistently (Line 216).

Point 10: Line 191: What are the units for “relative RNA expression? (relative to what?). “Contrast” = Control and “Add Cadmium” = Treatment?

Response 10: In the RT-PCR gene validation experiments, the levels of RNA expression observed in the same sample were influenced by the sample concentrations, the amplification, and so on. Therefore, an internal reference gene, which is stably expressed in each sample, is usually used to correct this kind of variation based on the standard procedure [7].

The ‘relative RNA expression’, which is the expression of a gene relative to the internal reference gene, is then used as an indicator of its expression in different samples [8,9]. The corresponding descriptions have been revised for better understanding (Lines 549–551).

In addition, “control" and "Cd-treated" were consistently used in the revised MS based on the reviewer’s suggestion (Line 260 and Line 279).

Qiu, H.; Durand, K.; Rabinovitch-Chable, H.; Rigaud, M.; Gazaille, V.; Clavere, P.; Sturtz, F.G. Gene expression of HIF-1alpha and XRCC4 measured in human samples by real-time RT-PCR using the sigmoidal curve-fitting method. Biotechniques 2007, 42, 355-362, doi:10.2144/000112331. Pfaffl, M.W. A new mathematical model for relative quantification in real-time RT-PCR. Nucleic Acids Res 2001, 29, e45, doi:10.1093/nar/29.9.e45. Pfaffl, M.W.; Horgan, G.W.; Dempfle, L. Relative expression software tool (REST) for group-wise comparison and statistical analysis of relative expression results in real-time PCR. Nucleic Acids Res 2002, 30, e36, doi:10.1093/nar/30.9.e36.

Point 11: transgenic?

Response 11: The corresponding descriptions of “transgenic Brassica campestris L.” and “transgenic hairy root” in the MS are inappropriate and have been revised to prevent misunderstanding. Please refer to Response 2 to reviewer 2 for details.

Point 12: Lines 213- 221: This is all general knowledge, not fit for the Discussion section.

Response 12: Thank you for your comments. This part has been substantially simplified for a better link to the following discussion according to the suggestion (Lines 283–287).

Point 13: Line 231: MDA experiments - what kind of experiments are that?

Response 13: Malondialdehyde (MDA) is one of the most frequently measured biomarkers of oxidative stress (e.g., lipid peroxidation). The corresponding descriptions in the MS were inappropriate and have been revised as follows: “In addition, a high Malondialdehyde (MDA) content and microscopic observation of the hairy roots indicated that cell membrane damage and apoptosis appeared, which increased our interests to further investigate the response molecular mechanism using RNA-Seq to perform transcriptome sequencing.” (Lines 287–291).

Point 14: Line 277: super enriched – enriched with what? Do you mean hyperaccumulating?

Response 14: Yes, this means hyperaccumulating. The term ‘hyperaccumulating/hyperaccumulator’ was consistently used in the revised MS (Lines 413–414).

Point 15: Line 289: toxicity of Barley – is barley a toxic plant?

Response 15: Thank you for your comments. This section was deleted because the Discussion section was rewritten.

Point 16: Line 318: phytochelins = phytochelatin

Response 16: Thank you for your comments. This error has been corrected in the revised MS (lines 226–227).

Point 17: Line 331: VsPCS1: You mean Phytochelatin Synthase 1 from Vicia sativa? (green arrow pea is Pisum sativum).

Response 17:VsPCS1’ is the abbreviation of Phytochelatin Synthase 1 from Vicia sativa. In the revised MS, the full name was directly used (Lines 412–413).

Point 18: Line 332: transferred – I guess you mean transduced?

Response 18: Yes, this was an inappropriate description and has been revised (Lines 410–411).

Point 19: Line 335 – 350: This is all recapitulation of results, you shouldn’t do that in the Discussion.

Response 19: Thank you for your comments. All recapitulation of results has been removed (Lines 453–458).

Point 20: Line 353: transgenic?

Response 20: The corresponding descriptions of “transgenic Brassica campestris L.” and “transgenic hairy root” in the MS were inappropriate and have been revised to prevent misunderstanding. Please refer to Response 2 to reviewer 2 for details.

Point 21: Line 356: Details are needed about the type of liquid medium and its composition.

Response 21: Thank you for your comments. The corresponding information has been added in this section (Lines 474–480).

Point 22: Line 430-437: These are not real conclusions, but recapitulations of your results.

Response 22: Thank you for your comments. The Conclusion section has been carefully revised and improved based on your suggestion (Lines 569–579).

Other changes in the MS

All changes in the revised manuscript have been marked in red. According to Suggestions of Reviewers/Editor, we used MPDI English editing service for our manuscript. Another grant number, 2017JBM073, was added due to support for the Article Processing Charges (APC) of this MS (Lines 593–596). The Introduction, Discussion, and Conclusions sections have been revised, and the literature references have been numerically re-listed in the Reference section. Due to the many changes in the article, we supplement the abbreviations (Line 599).

Round 2

Reviewer 2 Report

This manuscript was very thoroughly revised and the authors with commendable effort have dealt with all my earlier comments. The paper underwent a remarkable transformation. What I particularly liked is that in the revised version there is much more unity between problem statement, experiments and discussion. The links between mechanistic studies with hairy roots and the use of Brassica campestris for phytoremediation are now much more clear, which greatly adds to the value of the paper.

Only a few editorial issues remain.

Line 18: the metabolic process = metabolic processes

Line 21: The Western blot = A Western blot

In several places a space should be inserted after a full stop closing a sentence.

Line 54: “transgenic”: What was the modification here. Were the transcription factors overexpressed due to new promoters?

Line 100: maybe expand the heading of the table to read: “Summary of sequence reads for six RNA samples including three replicate control treatments (C1 – C3) and three samples of cadmium treatment (T1 – T3).

Line 102: “correlations of control groups” = “correlations among control replicates”; “correlations of Cd treatment groups” = “correlations among replicates within the Cd treatment”.

Line 103: I think it is important to add that correlations between control and treated samples were much lower.

Line 131: “Based on the results of the DEG detection “ – maybe better formulated as: “Following the identification of DEGs we analyzed their functions using GO.”

Line 172: Heading of table slightly expanded: “Table 2. Differential expression of eight target genes (relative to reference genes) quantified by qPCR in control and Cd-treated hairy root samples.”

Line 183: “The actin was stabilized.”= “The amount of actin protein was stable across the treatments”.

Line 184: “demonstrated the large-scale expression of GSHB and GST” = “demonstrated that not only the mRNA but also the proteins of GSHB and GST were highly upregulated.

Line 318-320: Please indicate which genes were used as reference genes in the qPCR. I assume it is beta-actin, but maybe there were more?

Line 327 -328: did you raise the antibodies yourself or were they purchased and if so, where.

Author Response

Response to Reviewer 1 Comments

Dear Reviewers:

Thank you very much for your letter and for editing the manuscript (MS) entitled “Comparative Transcriptome Analysis of the Molecular Mechanism of the Hairy Roots of Brassica campestris L. in Response to Cadmium Stress” (ID: ijms-632636). We would like to thank you for the time and effort spent reviewing our paper. Your comments are very helpful for improving the quality of our work. Motivated by the comments, we have carefully revised the manuscript and tried to fix all the problems you mentioned. All authors have seen the revised manuscript/response letter and approved to submit them to this excellent journal.

All revisions have been done using Word’s “track changes” function so that changes are easily visible. The revised MS has been uploaded for your review.

Below are our responses to the comments from all the reviewers. Thank you very much for your valuable time.

Sincerely,

Yan Qiong

Point 1: Line 18: the metabolic process = metabolic processes 

Response 1: Thank you for your comments. The description has been changed to " metabolic processes " (line 18).

Point 2: Line 21: The Western blot = A Western blot

Response 2: According to your kind comment, we have used " A Western blot " instead of " The Western blot " (line 21).

Point 3: In several places a space should be inserted after a full stop closing a sentence.

Response 3: Thank you for your comments. This error has been corrected in the revised MS (line 34, line 37, line 40).

Point 4: Line 54: “transgenic”: What was the modification here. Were the transcription factors overexpressed due to new promoters?

Response 4: The “transgenic plant” is “transgenic Arabidopsis” and we had been corrected it (lines 52 – 54). Yes, the transcription factors overexpressed due to new promoters [1].

Wu; Huilan; Chen; Chunlin; Du; Juan; Liu; Hongfei; Cui; Yan. Co-Overexpression FIT with AtbHLH38 or AtbHLH39 in Arabidopsis-Enhanced Cadmium Tolerance via Increased Cadmium Sequestration in Roots and Improved Iron Homeostasis of Shoots. Plant Physiology 2012, 158, 790.

Point 5: Line 100: maybe expand the heading of the table to read: “Summary of sequence reads for six RNA samples including three replicate control treatments (C1 – C3) and three samples of cadmium treatment (T1 – T3).

Response 5: Thank you for your comments, we have changed the description (lines 100 – 101).

Point 6: Line 102: “correlations of control groups” = “correlations among control replicates”; “correlations of Cd treatment groups” = “correlations among replicates within the Cd treatment”.

Response 6: Thank you for your comments. We have made corrections according to the Reviewer’s comment (lines 104 – 105)

Point 7: Line 103: I think it is important to add that correlations between control and treated samples were much lower.

Response 7: Thank you for your comments. In fact, we focused on the correlation within their respective groups. The high correlation indicated that the sample has good repeatability. The low correlation between control and treated samples indicated that the treatment caused changes in the transcription level of hairy roots. (line 106).

Point 8: Line 131: “Based on the results of the DEG detection “ – maybe better formulated as: “Following the identification of DEGs we analyzed their functions using GO.”

Response 8: Thank you for your comments. We have made corrections according to the Reviewer’s comment (lines 134 – 135)

Point 9: Line 172: Heading of table slightly expanded: “Table 2. Differential expression of eight target genes (relative to reference genes) quantified by qPCR in control and Cd-treated hairy root samples.”

Response 9: According to the Reviewer’s comment, we have changed in the revised MS (lines 176 – 177).

Point 10: Line 183: “The actin was stabilized.”= “The amount of actin protein was stable across the treatments”.

Response 10: We have made corrections according to the Reviewer’s comment (lines 188 – 189).

Point 11: Line 184: “demonstrated the large-scale expression of GSHB and GST” = “demonstrated that not only the mRNA but also the proteins of GSHB and GST were highly upregulated.

Response 11: According to the Reviewer’s comment, we have changed in the revised MS (lines 189 – 191).

Point 12: Line 318-320: Please indicate which genes were used as reference genes in the qPCR. I assume it is beta-actin, but maybe there were more?

Response 12: Previous studies indicated that probable ubiquitin-conjugating enzyme E2 21 (BnUBC21) gene was one of the top four choices as stably expressed reference genes for vegetative tissues [2]. Therefore, the UBC21 gene was used as a reference gene to correct gene expression (lines 328 – 330).

Chen, X.; Truksa, M.; Shah, S.; Weselake, R.J. A survey of quantitative real-time polymerase chain reaction internal reference genes for expression studies in Brassica napus. Anal Biochem 2010, 405, 138-140, doi:10.1016/j.ab.2010.05.032.

Point 13: Line 327 -328: did you raise the antibodies yourself or were they purchased and if so, where.

Response 13: According to the Reviewer’s comment, we have added the manufacturers’ information in the revised MS (lines 335 – 336).

Other changes in the MS

We added one more reference in the revised MS (line 329).